# Data-Driven Adaptive Control for Laser-Based Additive Manufacturing with Automatic Controller Tuning

**Lequn Chen [1,2]** , **Xiling Yao [1,*]** , **Youxiang Chew [1]** , **Fei Weng [1]** , **Seung Ki Moon [2,*]** and **Guijun Bi [1,*]**

1 Singapore Institute of Manufacturing Technology, Agency for Science, Technology and Research, 73 Nanyang Drive, Singapore 637662, Singapore; CHEN1189@e.ntu.edu.sg (L.C.); chewyx@simtech.a-star.edu.sg (Y.C.); weng_fei@simtech.a-star.edu.sg (F.W.)

2 School of Mechanical and Aerospace Engineering, Nanyang Technological University, 50 Nanyang Ave, Singapore 639798, Singapore

* Correspondence: yao_xiling@simtech.a-star.edu.sg (X.Y.); skmoon@ntu.edu.sg (S.K.M.); gjbi@simtech.a-star.edu.sg (G.B.)

**Abstract:** Closed-loop control is desirable in direct energy deposition (DED) to stabilize the process and improve the fabrication quality. Most existing DED controllers require system identifications by experiments to obtain plant models or layer-dependent adaptive control rules, and such processes are cumbersome and time-consuming. This paper proposes a novel data-driven adaptive control strategy to adjust laser voltage with the melt pool size feedback. A multitasking controller architecture is developed to incorporate an autotuning unit that optimizes controller parameters based on the DED process data automatically. Experimental validations show improvements in the geometric accuracy and melt pool consistency of controlled samples. The main advantage of the proposed controller is that it can adapt to DED processes with different part shapes, materials, tool paths, and process parameters without tweaking. System identification is not required even when process conditions are changed, which reduces the controller implementation time and cost for end-users.

**Keywords:** additive manufacturing; direct energy deposition; closed-loop control; virtual reference feedback tuning

## 1. Introduction

Laser-based direct energy deposition (DED) is an additive manufacturing (AM) process that is used to fabricate metallic components layer by layer and uses a laser as the heat source to melt additive materials (in either powder or wire form) as they are deposited onto a substrate [1]. The laser-based DED has found broad applications in the aerospace and marine industries due to its capability of making large-scale and customized parts in a cost-effective way. However, the laser-based DED process has poorer stability compared to traditional metal forming processes. It is prone to defects and dimensional inaccuracy due to various factors, including uneven thermal stress, strong melt pool dynamics, localized heat accumulation, inconsistent speed, and other unpredictable disturbances during laser beam delivery and material feeding [2]. Therefore, closed-loop control systems with sensor feedbacks are highly desirable for the laser-based DED process [3]. However, the nonlinearity and varying dynamics of the DED makes the search for robust control algorithms a challenging task. Constant controller parameters may not perform well for all the layers in DED, as the process conditions (e.g., the solidified layer temperature and the associated nominal melt pool size) change over time as the part grows [4]. Moreover, a controller designed for a specific material may not be

suitable for another material, since the nominal DED process parameters and material properties (e.g., thermal conductivity, viscosity, and emissivity) are different. Therefore, this research aimed to develop a data-driven adaptive control method in which the controller parameters are variables that can be automatically updated during the laser-based DED process.

Melt pool characteristics have a strong correlation with the process stability and part quality in DED, and hence they are frequently used in closed-loop control systems [5]. The effects of DED process parameters on melt pool characteristics have been investigated quantitatively in previous research. It was found that the melt pool size and temperature are both positively influenced by the input energy density and hence the laser power [6,7]. The proportional–integral–derivative (PID) controller has been widely adopted in the development of melt-pool-based DED control systems due to its simplicity and effectiveness. For example, Bi et al. [8,9] used a pyrometer to sense the infrared (IR) radiation from the melt pool and sent the reading to a PID controller. Based on the error term calculated as the difference between the real-time IR signal and its nominal value, the PID controller could change the laser power in response to the fluctuation in melt pool temperature. Consistent height of the as-built part was achieved by the above approach. Hofman et al. [10] utilized a complementary metal-oxide-semiconductor (CMOS) camera to capture the melt pool image and measure the melt pool width. Then, they applied a PID controller to increase and decrease the laser power to compensate for the rise and fall of the melt pool width, respectively. Similar studies used PID controllers to adjust the laser power based on the variation of melt pool area [11,12]. Enhancements have also been made to the conventional PID method, aiming to improve the controller performance under the varying dynamics of the DED process. For example, Moralejo et al. [13] added a feedforward path to a PID controller, which could reduce the overshoot and improve the response speed. The authors also embedded the melt pool size setpoint into the preprogrammed computer numerical control (CNC) code. The position-dependent setpoint allowed the building of changeable geometries using a single-track toolpath. However, extensive experimentation was needed to obtain the correct controller parameters. Akbari and Kovacevic [4] implemented an adaptive control strategy that handled the variation of melt pool response across multiple layers. System identification was performed for each layer, and the response of melt pool size to the laser power was represented by a first-order transfer function that had different coefficients for different layers. A PID controller was used to adjust the laser power; but instead of having constant parameters, its PID parameters were tuned for each layer using the corresponding transfer function. This strategy allowed the controller to be adaptable to changes in the heat conduction mode and cooling rate as the part height increased. However, the layer-by-layer system identification and controller tuning process was time-consuming and lacked automation, which made the above control strategy less user-friendly for industry applications. Song et al. [14] proposed a two-input single-output hybrid controller that consisted of a rule-based height controller and a closed-loop temperature controller. The laser power was reduced by the height controller until the melt pool height was below the preset layer thickness threshold. Afterward, the temperature controller took over and adjusted the laser power based on the pyrometer feedback. Another hybrid control strategy was proposed for a laser–wire DED system by Gibson et al. [15]. The laser power was controlled by the melt pool geometry using thermal camera feedback, while the printing speed and wire feeding rate were controlled by the part height on a per-layer basis. As the part height increased, both the printing speed and wire feeding rate increased. Hence, the average laser energy density decreased, which was allowed due to the heat accumulation in the freshly built layers below the melt pool. This approach could maintain the process stability and improve the productivity at the same time. However, since the heat accumulation effect strongly depended on the material, geometry, area, and maximum height of the part, the selection of controller parameters for a specific product might not be suitable for another product.

One of the limitations in the existing control strategies that extensive experimentation is required to find the optimal controller parameters. The system identification and parameter tuning processes are cumbersome and time-consuming. Besides, although some of the aforementioned controllers

considered interlayer changes in DED process conditions, they did not adapt to the intralayer variation of melt pool dynamics. Therefore, in this research, we propose a novel data-driven adaptive control strategy with automatic parameter tuning instead of using a prior plant model or static controller parameters. During the laser-based DED process, the sensor-captured melt pool size and the laser voltage signal are recorded in each time frame as the system input and output (I/O) data, respectively. The I/O data collected within a periodic time interval are stored in a buffer before they are fed into an autotuning unit to compute the optimal PID controller parameters. The PID controller with the updated parameters is used to adjust the laser power in the next time interval while the new sets of I/O data are being collected to overwrite the buffer. The controller parameters are reoptimized once again at the end of the cycle, using the updated I/O data that reflect the varying response dynamics of the melt pool. The virtual reference feedback tuning (VRFT) algorithm [16] is implemented in the autotuning unit for PID parameter optimization. The data-driven controller update is performed periodically throughout the entire DED process regardless of the present time, layer, material, size, or shape. Prior and interlayer system identification experiments are no longer needed, thus saving time and cost. Besides, since the proposed adaptive controller is dynamically set by the time-dependent process data, it can be applied to parts with any materials, geometries, and sizes without modification, which makes its adoption convenient for industry end-users.

This paper is organized as follows: Section 2 introduces the overall setup of the laser-based DED system with melt pool monitoring and closed-loop control. Section 3 explains the details of the proposed data-driven adaptive control strategy. Section 4 presents the experimental results that demonstrate the effectiveness of the proposed method. Lastly, Section 5 concludes the paper and provides direction for future research.

## 2. System Setup

This research was conducted on an in-house-developed laser-based DED system. Figure 1 shows a simplified illustration of the system setup, where the transmissions of energy and signal are represented by solid arrows. A six-axis IRB-4400 industrial robot (ABB, Zürich, Switzerland) carried the optical head and a two-axis IRBP-A positioner (ABB, Zürich, Switzerland) held the substrate. The laser beam with 1070 nm wavelength was supplied by a YLS-6000 Ytterbium laser source (IPG Photonics, Oxford, MA, USA) with the maximum power of 6 kW. A BIMO optical head (HIGHYAG, Kleinmachnow, Germany) received the laser beam via fiber and focused the beam onto the substrate as it melted the metal powders. A powder feeder (GTV, Luckenbach, Germany) was used to deliver the metallic powder to the nozzle installed at the bottom of the optical head. A WAT-902B charged-couple device (CCD) camera (Watec, NY, USA) was mounted on the optical head. Through a series of reflective optics, the melt pool image could be captured by the CCD camera coaxially. The viewing direction was perpendicular to the melt pool that was located at the center of the camera view. The melt pool emits a larger amount of near-infrared (NIR) radiation than its surroundings due to its higher temperature. Therefore, a NIR band-pass filter with a bandwidth of 780–1000 nm was attached to the CCD camera so that the melt pool could be isolated from the surroundings without sensing the diffusively reflected 1070 nm laser.

A personal computer (PC) running an Ubuntu 18.04 LTS operating system was used as the main controller. It was responsible for sensor data collection, image processing, and control algorithm execution. The output channel of the CCD camera was connected to the controlling PC that received the digital image data via a USB 3.0 port. The raw image in grey-scale pixels was processed by a series of computer vision algorithms using the OpenCV library [17]. The melt pool area was cropped from the raw image by a circular mask that has a diameter slightly smaller than that of the nozzle outlet so that the NIR light reflected by the nozzle's inner surface could be removed. Then, a filter with a prescribed threshold was applied to binarize the melt pool image, after which an ellipse was fit into the binary image, as shown in Figure 1. The melt pool width (MPW) was approximated by the minor axis of the resulting ellipse. The MPW is influenced by the quality of interlayer fusion and heat transfer

mode, and it is an indicator of the part integrity and surface roughness of DED-fabricated parts [6]. Therefore, the MPW value was sent to the controller as the feedback data. The proposed data-driven adaptive controller was implemented in an in-house-developed software program running on the PC. The output of the controller was the analog voltage signal supplied to the laser source. The laser voltage ranging from 0 to 10 V determined the actual laser power. The digital on/off signal of the laser emission was sent from the robot's control box to the laser source via hardwiring, which did not interfere with the laser voltage sent from the PC. The computation of the output laser voltage based on the feedback melt pool data using the proposed control strategy is discussed in the next section.

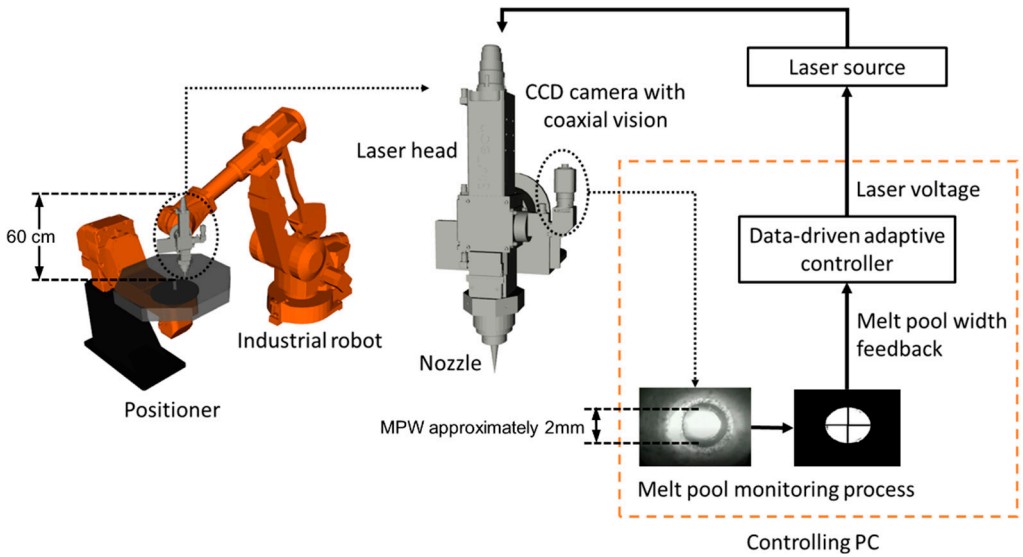

**Figure 1.** The setup of the laser-based direct energy deposition (DED) system with closed-loop control.

## 3. The Data-Driven Adaptive Control Strategy

### 3.1. Conventional Proportional–Integral–Derivative (PID) Algorithm

This section introduces the formulation of a conventional PID controller and its parameter optimization problem, which lays the foundation of the proposed data-driven adaptive control strategy. In the DED process, the goal of closed-loop control is to improve the stability of the MPW by adjusting the laser voltage that determines the laser power. The block diagram for the conventional closed-loop control system is shown in Figure 2. The PID control action in the continuous time-domain can be expressed as:

$$u(t) = K_p e(t) + K_i \int e(t)dt + K_d \frac{d}{dt}e(t) \tag{1}$$

where $e(t)$ is the error term that equals to the difference between the reference MPW value ($r(t)$) and the measured MPW value ($y(t)$); $K_p$, $K_i$, and $K_d$ are the tuneable PID parameters; and $u(t)$ is the laser voltage signal computed by the PID controller. The laser voltage signal is input into the plant $G(s)$ that produces the MPW feedback. The PID controller with a first-order low-pass filter in derivate term can be represented as the following s-domain transfer function [18]:

$$C(s) = K_p + K_i \frac{1}{s} + K_d \frac{s}{1 + \tau_d s} \tag{2}$$

where $\tau_d$ is the first-order derivative filter time and $s$ is the complex variable in the frequency domain.

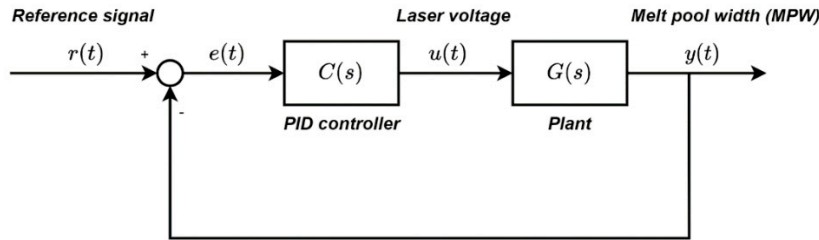

**Figure 2.** Block diagram of the conventional closed-loop control system for the direct energy deposition (DED) process.

To obtain the optimal controller parameters denoted as $\theta = (K_p, K_i, K_d)$, the following optimization problem in the model-reference (MR) framework [19] can be established as follows:

$$\hat{\theta}_{MR} = \arg\min_{\theta} J_{MR}(\theta) \tag{3}$$

$$J_{MR}(\theta) \triangleq \left\| \left( \frac{G(z^{-1})}{1 + G(z^{-1})C(z^{-1};\theta)} - T_d(z^{-1}) \right) L_r(z^{-1}) \right\|_2^2 \tag{4}$$

where the cost function $J_{MR}(\theta)$ penalizes the difference between the desired closed-loop transfer function $T_d(z^{-1})$ and the actual closed-loop transfer function, $G(z^{-1})$ and $C(z^{-1};\theta)$ are the discrete time-domain counterparts of $G(s)$ and $C(s)$, respectively, and $L_r(z^{-1})$ is a band-pass noise filter. The optimization objective is to minimize the $J_{MR}(\theta)$ criterion. The controller $C_d(z^{-1})$ with the optimal parameters $\hat{\theta}_{MR}$ is the final tuning outcome, and the resultant closed-loop transfer function should be equal to the desired closed-loop reference model $T_d(z^{-1})$, i.e.,

$$\frac{G(z^{-1})}{1 + G(z^{-1})C_d(z^{-1};\theta)} = T_d(z^{-1}) \tag{5}$$

Since the plant model $G(z^{-1})$ in the cost function $J_{MR}(\theta)$ is unknown, system identification is needed in the conventional tuning process to find $G(z^{-1})$ before controller parameters can be optimized; otherwise, trial-and-error experiments are conducted to determine the PID gains, which is cumbersome and time-consuming. The inaccuracy in the system identification could also jeopardize the controller tuning result. Moreover, for different part geometries and powder materials, the DED process does not have a single plant model $G(s)$ that can generalize the melt pool dynamic response to the laser voltage signal. The melt pool response also varies with time when fabricating the same part, and hence a single set of optimal controller parameters $\theta$ cannot be obtained for the entire time domain. Therefore, an adaptive control method is needed to automatically update the controller parameters without repeated system identification. The proposed data-driven adaptive controller is explained in Section 3.2.

### 3.2. Adaptive Controller Design

The proposed adaptive controller is implemented with a multitasking architecture, as illustrated in Figure 3. Three main tasks are executed concurrently during the DED process, i.e., the melt pool monitoring unit, the autotuning unit, and the digital PID unit. Each of them contains subtasks performing different functions automatically and continuously. Data transmission within the controller is indicated by the dotted lines in Figure 3, and the controller parameter update routine is highlighted by the blue line. Details of the above three units are given below.

In the melt pool monitoring unit, the "camera driver" subtask reads the raw melt pool image captured by the co-axial CCD camera, and the "image processing" subtask performs the masking, binarization, and ellipse fitting procedures as described in Section 2. The minor axis length of the fitted

ellipse, measured in the number of pixels, is published as the MPW data. The MPW data are received by both the digital PID unit and the autotuning unit.

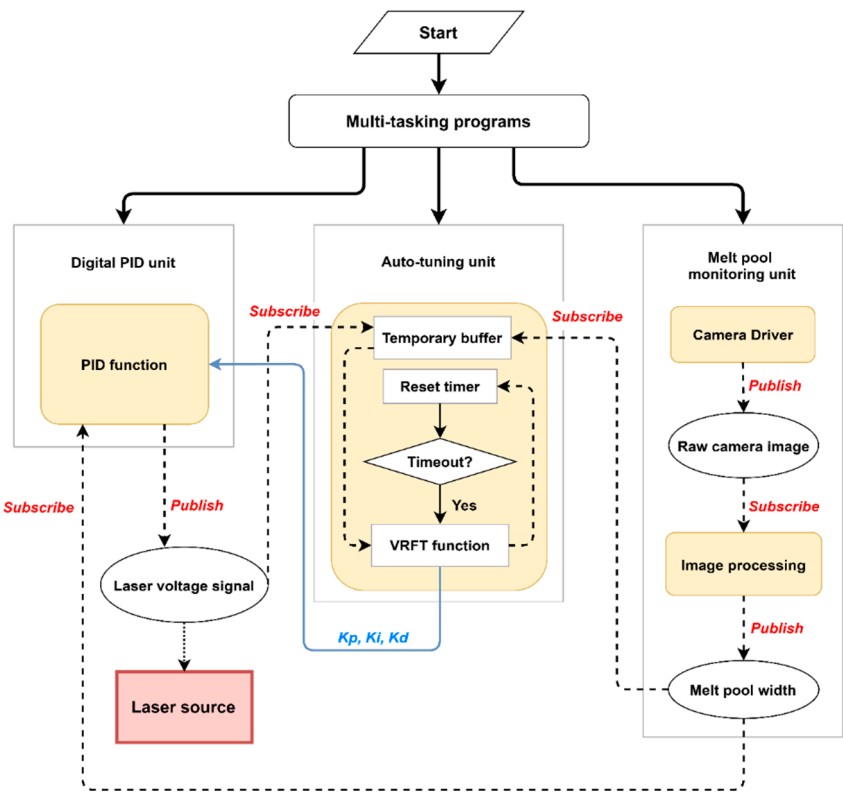

**Figure 3.** The multitasking architecture of the proposed data-driven adaptive controller.

The digital PID unit calculates the output laser voltage based on the MPW feedback using the standard formulations in Equations (1) and (2). However, instead of using predetermined and user-specified constant controller parameters, this PID unit accepts the adaptive parameters $\left(K_p,\ K_i,\ K_d\right)$ calculated in situ by the autotuning unit. System identification by experimental trials-and-errors is removed from the controller design procedure, and the prior knowledge of a plant model is no longer required. The output laser voltage is sent to the laser source as an analog signal, and at the same time it is subscribed by the autotuning unit as an input to update the $K_p$, $K_i$, and $K_d$ parameters.

The autotuning unit has two responsibilities, i.e., (1) collecting the process data generated by the other two units and (2) using the process data to update the controller parameters repeatedly and automatically, thus achieving the data-driven adaptive control capability. The autotuning unit consists of a temporary data buffer, a timer function, and a VRFT function. The MPW and laser voltage are recorded in each time frame as the system input and output (I/O) data, respectively. The I/O data collected within a periodic time interval are stored in the temporary buffer before they are extracted by the VRFT function. The timer function launches the VRFT function when the timeout signal is issued. The VRFT function computes the optimal controller parameters using the I/O data in the buffer and sends the updated $K_p$, $K_i$, and $K_d$ values back to the digital PID unit. The timer is reset upon completion of the VRFT routine, and the temporary buffer is flushed. The PID unit with the updated parameters adjusts the laser voltage in the next time interval while the new sets of I/O data are being collected by the buffer. Periodically, the controller parameters are reoptimized at the end of each timer cycle using the updated I/O data until the end of the DED process.

Figure 4 shows the block diagram of the proposed adaptive controller. In addition to the error feedback in the conventional PID controller (Figure 2), the autotuning unit forms the second feedback loop that updates the $K_p$, $K_i$, and $K_d$ parameters automatically. The reference model, denoted by $T_d(s)$,

is an s-domain transfer function representing the desired closed-loop system behavior (e.g., desired settling time and desired response speed) [20], and it can be written as follows:

$$T_d(s) = \frac{e^{-s\tau}}{(1+0.2t_s s)^n} \tag{6}$$

where the settling time $t_s = 0.01$ s, the response delay time $\tau = 0$, and $n = 1$ are specified for the stable steady-state tracking purpose [20]. The reference model $T_d(z^{-1})$ in Figure 4 is the z-domain transfer function that is computed by transforming $T_d(s)$ into the discrete-time domain using the bilinear method [21].

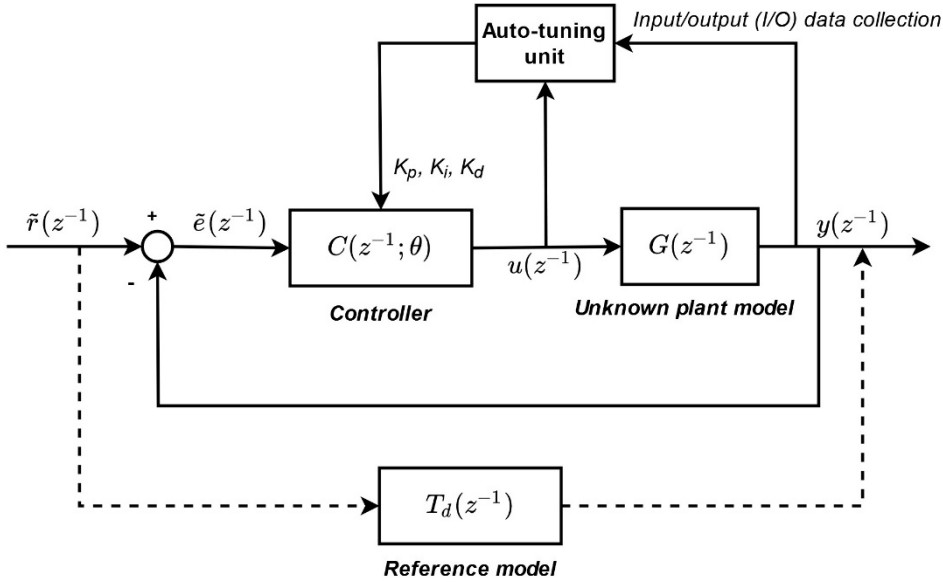

**Figure 4.** Block diagram of the adaptive controller with an autotuning unit.

The objective of the VRFT function in the autotuning unit is to optimize the controller so that the resulting closed-loop transfer function is identical to the reference model $T_d(z^{-1})$. Only the I/O data collected during the experiment are used directly in the VRFT function. The plant model $G(z^{-1})$ can remain unknown since it is not required in the VRFT function, and hence the system identification can be eliminated from the controller tuning process. More details of the fundamental VRFT theories can be found in [16,22]. The reference model $T_d(z^{-1})$ generates the desired system output $y_d(z^{-1})$, which is expressed as

$$y_d(z^{-1}) = T_d(z^{-1})r(z^{-1}) \tag{7}$$

where $r(z^{-1})$ is the reference signal (i.e., the setpoint). During the laser-based DED process, the laser voltage signal and MPW data collected during the time interval $\Delta t$ are denoted by

$$D = \{(u_k,\ y_k),\ k = 1,2\ldots N\} \tag{8}$$

where $u_k$ and $y_k$ are the $k$th instance of the laser voltage and MPW data, respectively. Given the reference model $T_d(z^{-1})$ and system output $y(z^{-1})$, the virtual reference signal can be computed as

$$\widetilde{r}(z^{-1}) = T_d^{-1}(z^{-1})y(z^{-1}) \tag{9}$$

where $y(z^{-1})$ is the discrete-time transform of the measured MPW dataset $\{y_1, y_2, y_3, \ldots y_N\}$. The virtual reference signal $\widetilde{r}(z^{-1})$ is not the actual input signal used to generate the resulting MPW $y(z^{-1})$. Instead,

it is the desired reference signal fed into the reference model $T_d(z^{-1})$ if we consider the measured output $y(z^{-1})$ as the desired system output $y_d(z^{-1})$.

Based on the computed virtual reference signal $\widetilde{r}(z^{-1})$, the virtual error signal $\widetilde{e}(z^{-1})$ is defined as the difference between the virtual reference signal and the measured system output, i.e.,

$$\widetilde{e}(z^{-1}) = \widetilde{r}(z^{-1}) - y(z^{-1}) = \left(T_d^{-1}(z^{-1}) - 1\right)y(z^{-1}) \tag{10}$$

The unknown plant model $G(z^{-1})$ can produce the MPW $y(z^{-1})$ when it is fed with the measured laser voltage $u(z^{-1})$. Therefore, we can define an "ideal controller" $C(z^{-1}, \theta)$ that should generate the measured laser voltage $u(z^{-1})$ when it is fed with the virtual error signal $\widetilde{e}(z^{-1})$. The above control action can be written as

$$u(z^{-1}) = C(z^{-1}, \theta)\widetilde{e}(z^{-1}) \tag{11}$$

The controller $C(z^{-1}, \theta)$ can be represented in the general format of:

$$C(z^{-1}, \theta) = \rho^T(z^{-1})\theta \tag{12}$$

where $\theta$ is the variable controller parameter and $\rho^T(z^{-1})$ is a vector of transfer functions. For a discrete PID controller $C_{pid}(z^{-1}, \theta)$, the following expression can be obtained by applying the bilinear transform to Equation (2), and the $\theta$ and $\rho(z^{-1})$ terms in Equation (12) are now $\begin{bmatrix} K_p & K_i & K_d \end{bmatrix}$ and $\begin{bmatrix} 1 & \frac{T_s}{2}\frac{1+z^{-1}}{1-z^{-1}} & \frac{2}{T_s}\frac{1-z^{-1}}{3-z^{-1}} \end{bmatrix}^T$, respectively.

$$C_{pid}(z^{-1}, \theta) = \begin{bmatrix} K_p & K_i & K_d \end{bmatrix} \begin{bmatrix} 1 \\ \frac{T_s}{2}\frac{1+z^{-1}}{1-z^{-1}} \\ \frac{2}{T_s}\frac{1-z^{-1}}{3-z^{-1}} \end{bmatrix} \tag{13}$$

In order to find the "ideal controller" that generates the laser voltage signal $u(z^{-1})$ with the feedback error $\widetilde{e}(z^{-1})$, the following optimization problem is solved in the VRFT function:

$$\hat{\theta}_{VRFT} = \arg \min_{\theta} J_{VRFT}(\theta) \tag{14}$$

$$J_{VRFT}(\theta) \triangleq \| u(z^{-1}) - C(z^{-1}; \theta)\widetilde{e}(z^{-1})\|_2^2 = \frac{1}{N}\sum_{k=1}^{N}\left\{u_k - C(z^{-1}; \theta)\left(T_d^{-1}(z^{-1}) - 1\right)y_k\right\}^2 \tag{15}$$

The above optimization problem can be solved by the quadratic programming (QP) method that searches for the best controller parameter $\hat{\theta}_{VRFT}$ to minimize the $J_{VRFT}(\theta)$ criterion. In this research, the solving algorithm was implemented in Python, adopted from the Pyvrft library [23]. The VRFT optimization problem is a mathematical equivalence to the conventional model-reference optimization problem described by Equations (3) and (4), as proven in [22]. A fixed plant model $G(z^{-1})$ required in Equations (3) and (4) is no longer included in Equations (14) and (15), which contributes to the adaptability of the proposed data-driven controller. The filtered error signal $\widetilde{e}_F(k)$ and laser voltage signal $u_F(k)$ can be expressed as follows:

$$\widetilde{e}_F(k) = L(z^{-1})\widetilde{e}(k), \quad u_F(k) = L(z^{-1})u(k) \tag{16}$$

The above-filtered signals are then used in the formulation of a modified cost function $J_{VRFT}(\theta)$ as follows:

$$
\begin{aligned}
J_{VRFT}(\theta) &= \tfrac{1}{N} \sum_{k=1}^{N} \left\{ u_F(k) - C(z^{-1};\theta)\widetilde{e}_F(k) \right\}^2 \\
&= \tfrac{1}{N} \sum_{k=1}^{N} \left\{ L(z^{-1}) \left[ u(k) - C(z^{-1};\theta)\widetilde{e}(k) \right] \right\}^2 \\
&= \tfrac{1}{N} \sum_{k=1}^{N} \left\{ L(z^{-1}) \left[ u(k) - C(z^{-1};\theta)(T_d^{-1}(z^{-1}) - 1)y(k) \right] \right\}^2
\end{aligned}
\tag{17}
$$

The filter $L(z^{-1})$ is formulated as follows, which makes the resultant PID controller a good approximation of the "ideal controller" [16]:

$$
L(z^{-1}) = \frac{(1 - T_d(z^{-1}))T_d(z^{-1})}{\Phi_u(z^{-1})^{1/2}}
\tag{18}
$$

The $\Phi_u(z^{-1})$ term in Equation (18) is the spectral density of the laser voltage signal, which can be calculated based on the I/O dataset $\{u(k)\}_{k=1,2\ldots N}$ by periodogram and ARMA modeling methods [24,25].

The execution of the VRFT-based autotuning unit in the laser-based DED process can be summarized in the following steps:

Step 0 **VRFT presetting**: Before the control process starts, the reference model $T_d(z^{-1})$, representing the desired system performance, is specified by Equation (6).

Step 1 **Data collection and preprocessing**: During the *M*th adaptive control cycle, the laser voltage and MPW data $D_M = \{(u_k, y_k)\}_M$ are recorded and stored into the temporary buffer within the interval $\Delta t = t_0 \sim t_N$. The virtual reference signal $\widetilde{r}_M(z^{-1})$ and the virtual error signal $\widetilde{e}_M(z^{-1})$ for this cycle are then calculated using Equations (9) and (10). The filter $L(z^{-1})$ determined by Equation (18) is applied to filter the I/O data and virtual signals.

Step 2 **VRFT controller tuning**: When the *M*th control cycle has completed, and the timeout signal is issued in the autotuning unit, the VRFT function updates the optimal controller parameters $\hat{\theta}_M$ by minimizing the modified cost function $J_{VRFT}(\theta)$ in Equation (17), using the data $D_M = \{(u_k, y_k)\}_M$ in the buffer.

Step 3 **PID parameter updating**: The autotuning unit sends the updated controller parameters $\hat{\theta}_M$ to the digital PID controller unit, resets the timer, flushes the buffer, and then returns to Step 1 to start the (*M+1*)th control cycle.

The main contribution of the proposed DED control strategy is that the autotuning method has eliminated the necessity of prior system identification and its associated cost and manual labor. The same controller can be applied in the DED fabrication with any shape, size, or material without modification. Previous rule-based adaptive DED control methods updated the controller parameters based on the layer number or the part height, while the rules were derived from experiments for a specific combination of material and process parameters [4,14,15]. In comparison, the proposed data-driven adaptive controller can update the parameters automatically regardless of the layer number or part height, and no prior experiments are needed to generate the control rules, thus saving time and cost for end-users.

## 4. Experimental Validation

The proposed data-driven adaptive controller was implemented in the laser-based DED system and validated experimentally. As listed in Table 1, different materials, geometries, and deposition tool paths were tested to validate that the proposed controller could automatically adjust its parameters and was adaptable to different deposition situations without the necessity to conduct system identification. Specifically, three experiments were conducted: (1) solid semicylinder with profile tool paths in 316 L stainless steel, (2) solid semicylinder without profile tool paths in 316 L stainless steel, and (3) thin-wall

pipe with a continuous spiral tool path in LPW-35N nickel alloy (a customized material designed by the authors' organization). Experiments 1 and 2 each comprised three samples. The first sample was fabricated using the constant nominal process parameters listed without the employed control system in Table 2. The second sample was fabricated using a conventional PID controller, where constant PID gains were used. The PID gains were determined by experiment-based system identification and trial-and-error. The third sample was fabricated using the proposed adaptive control method, where PID gains were automatically optimized and updated during the process. These three samples were compared with each other, and the effect of the adaptive controller was analyzed. Experiment 3 was conducted to validate that the proposed adaptive control method was still effective even when the powder material and part geometry were changed (compared to Experiments 1 and 2), while no additional experiment was needed to recalibrate the controller. The MPW data and the corresponding laser voltage output in these three experiments were recorded, and the results are discussed below.

**Table 1.** Materials, geometries, and deposition tool paths in different experiments.

| Experiment Number | Powder Material | Geometry | Deposition Tool Path |
|:---:|:---:|:---:|:---:|
| 1 | 316 L stainless steel | Solid semicylinder | Zigzag infill with the profile tool path |
| 2 | 316 L stainless steel | Solid semicylinder | Zigzag infill without the profile tool path |
| 3 | LPW-35N nickel alloy | Thin-walled hollow pipe | Continuous spiral single-bead tool path |

**Table 2.** Constant nominal process parameters used for direct energy deposition (DED) processes without control.

| Process Parameters | Powder Material | | Unit |
|:---:|:---:|:---:|:---:|
| | **316 L Stainless Steel** | **LPW-35N Nickel Alloy** | |
| Laser voltage | 6.2 | 6.2 | V |
| Printing speed | 20.0 | 20.0 | mm/s |
| Powder feeding rate | 6.09 | 7.73 | g/min |
| Layer thickness | 0.2 | 0.3 | mm |
| Infill hatch distance (for solid parts only) | 2.0 | 2.0 | mm |

Figure 5 shows the results of Experiment 1, comparing the samples of depositing the 30 mm diameter semicylinder structure using 316 L stainless steel. The part's nominal height ($H_N$) was 9 mm, and the nominal semicylinder diameter ($D_N$) was 30 mm, as indicated in Figure 5. In Figure 5a, the semicylinder structure was fabricated using the constant laser voltage signal (6.2 V) without control. A laser profiler was used to scan the surface of the part and produce its 3D point cloud using the method introduced in [26,27]. The scanned surface reconstructed from the point cloud is shown in Figure 5b, where the color bar indicates the distance in the z-direction from the points to the reference plane [26]. The larger distance shown in the graph means the lower dimensional accuracy of the DED-fabricated part. A bulging area at the center of the surface that is significantly higher than the edges can be seen in Figure 5b, which was caused by the unstable laser energy density and hence the uneven heat accumulation in the part. Figure 5c shows the sample deposited using a conventional PID controller with constant PID gains (($K_P$, $K_I$, $K_D$) = (0.04, 0.02, 0.00)). The PID gains were tuned based on Ziegler-Nichols criteria [28] via trial-and-error experiments, which was considerably time-consuming and ineffective in terms of manpower and cost. The top surface of the conventional PID controlled sample is shown in Figure 5d. It can be seen that its surface was flatter and the central bulge height was smaller than that in the uncontrolled sample. This is due to the effect of the closed-loop control on stabilizing the heat input and reducing the localized heat accumulation. Figure 5e shows the sample deposited with the proposed adaptive controller, and its surface was

also scanned and reconstructed, as shown in Figure 5f. It can be observed that the central bulge area was further flattened compared with the conventional PID controlled sample. Figure 6a–c shows the time plots of the MPW data and laser voltage signals for the uncontrolled sample, conventional PID controlled sample, and adaptively controlled sample, respectively. In Figure 6a, a progressive increase in the MPW is observed due to the accumulated heat built up in the part when using a constant laser voltage. The MPW exceeded 140 pixels at the end of the fabrication process with the trend of growing even larger. In Figure 6b, the conventional PID controller was used, and the growth of MPW was reduced compared to the uncontrolled sample. However, there was still an increasing trend in MPW. About half of the MPW data were in the range of 120–130 pixels after 250 s, suggesting an unstable heat input across the surface and the possibility of more severe geometry inaccuracy if the deposition continues. In Figure 6c, the adaptive controller showed a more significant effect on stabilizing the melt pool than the conventional PID controller. The laser voltage signal was reduced gradually to decrease the average energy density as the process continued. The MPW data were maintained at a narrower range of 115–120 pixels throughout the entire deposition process. Therefore, uneven localized heat accumulation was minimized, and the possibility of surface defect occurrence was reduced.

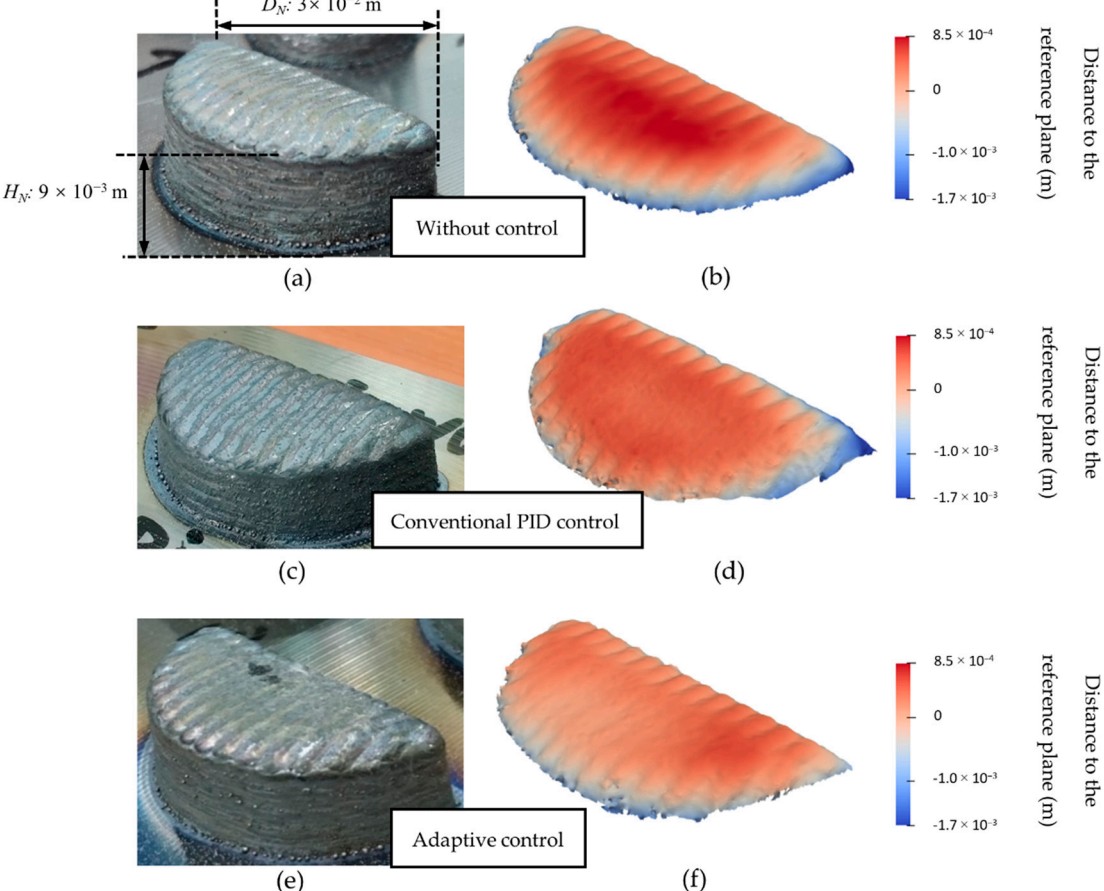

**Figure 5.** Samples of Experiment 1 (solid semicylinder with the profile tool path, 316 L stainless steel). (**a**) The sample fabricated without control; (**b**) The reconstructed surface of the uncontrolled sample; (**c**) The sample fabricated with a conventional proportional–integral–derivative (PID) controller; (**d**) The reconstructed surface of the conventional PID controlled sample; (**e**) The sample fabricated with the proposed adaptive controller; (**f**) The reconstructed surface of the adaptively controlled sample.

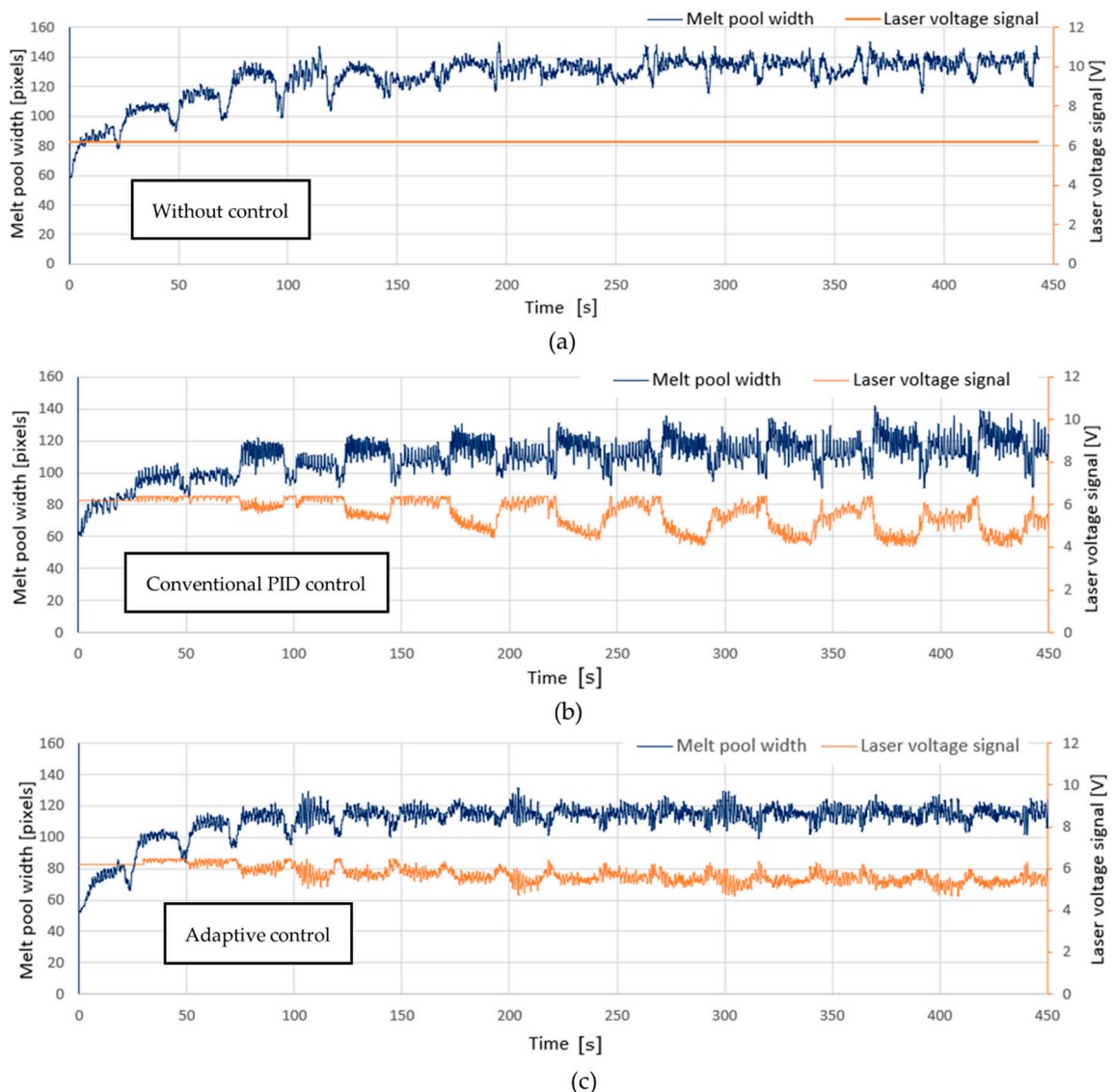

**Figure 6.** Time plots of the melt pool width (MPW) and laser voltage signals for Experiment 1 (solid semicylinder with the profile tool path, 316 L stainless steel). (**a**) The data without control; (**b**) The data with conventional PID control; (**c**) The data with adaptive control.

During the closed-loop control, the laser voltage output was constrained by a maximum value of 6.2 V. This value was the limit of the DED process window determined in previous experiments, in order to prevent material failure. In fact, as shown in Figure 6b,c, the laser voltage was below 6.0 V for most of the time and seldomly hit the 6.2 V limit.

Figure 7 shows the PID gain variations for the entire time-plots extracted from Figure 6c, where the controller parameters ($K_P$, $K_I$, $K_D$) were updated in each adaptive control interval (10 s in this study). The collected I/O dataset in each interval was used to optimize the controller parameters automatically without manual tuning. The resultant controller parameters were truncated at three decimals, as shown in the plot. The resultant PID controller had zero derivative gains ($K_D = 0$), which was acceptable in this case since the negligible derivative action could reduce the system's sensitivity to noises. When the first layer was deposited, a significant portion of the input heat was conducted to the substrate, and hence the melt pool was in a transient state. Since neither surface defect nor geometric nonconformance was observed in the first layer, a constant laser voltage was applied. As the deposition process continued, the heat transfer rate between part and substrate reached a steady state, and the local fluctuation of the

MPW would potentially lead to geometric inaccuracies. Therefore, the control action was started in the second layer at the time around 30 s. After the second layer, the MPW was stabilized and maintained within 115–120 pixels.

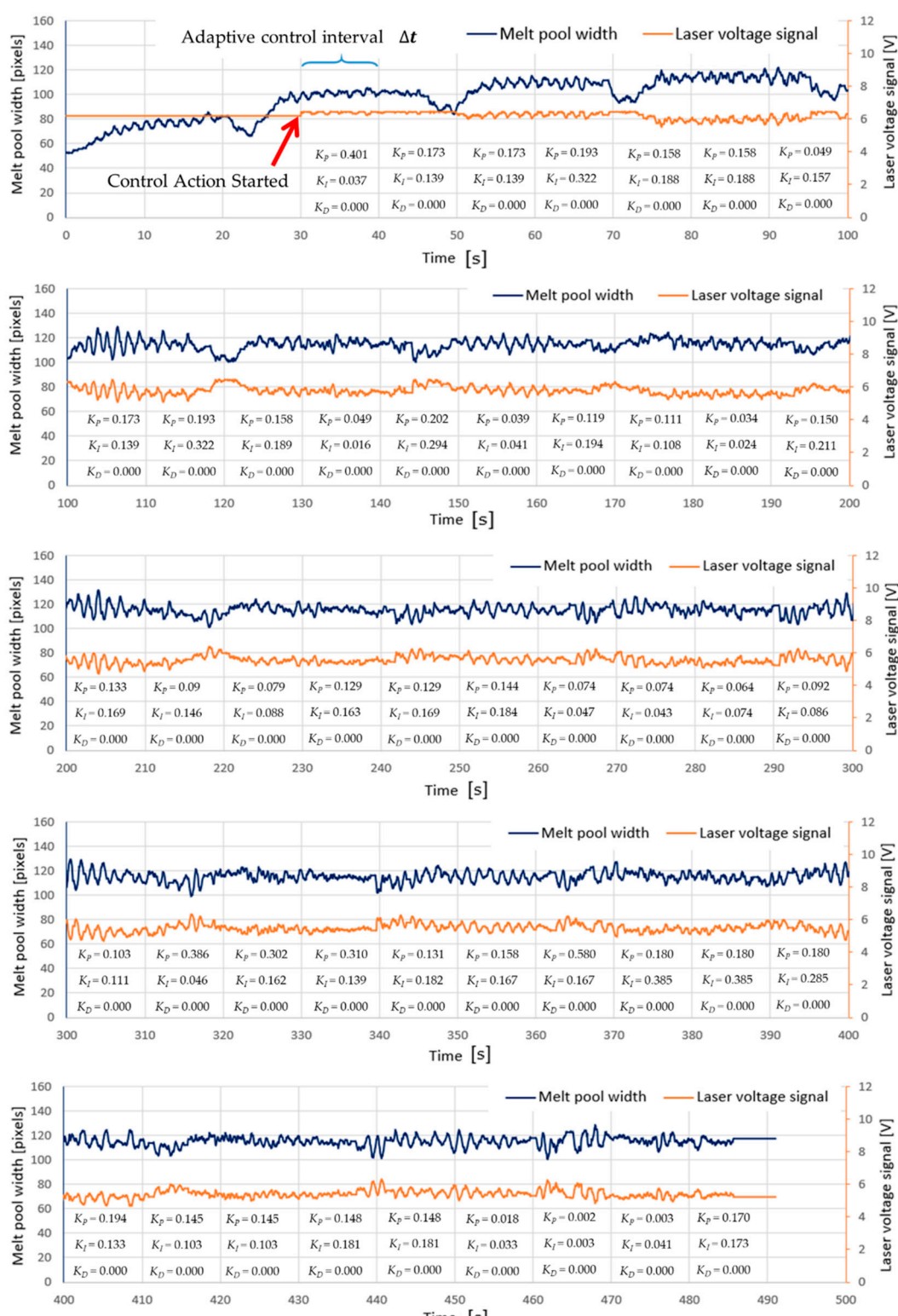

**Figure 7.** The MPW data for the entire time-domain extracted from Figure 6c, showing the automatic update of the controller parameters ($K_P$, $K_I$, $K_D$).

Figure 8 illustrates the results of Experiment 2. The samples in Experiment 2 had the same nominal dimensions as those in Experiment 1 but with different tool path settings. In particular, the samples printed in Experiment 2 did not include the profile tool path. Only the infill was built, while the outer contour of each layer was not deposited. Figure 8a,b shows the sample fabricated with the constant laser voltage (6.2 V) without control. With a higher central bulge and lower edges, this sample had a more significant distortion than that seen in Figure 5 in Experiment 1, which was the result of less material added to the edges to compensate for the distortion when the profile tool path was absent. The conventional PID control method was used for the sample in Figure 8c,d. Both the height and area of the surface bulge area were reduced compared to those in the uncontrolled sample. However, before the PID controller was deployed, system identifications and trial-and-error experiments were conducted to determine the PID gains, thus introducing extra time and material wastage. With the proposed adaptive controller employed, as shown in Figure 8e,f, the sample had a flatter surface, sharper edges, and hence a better geometric accuracy than its uncontrolled and conventional PID controlled counterparts. Figure 9 shows the time plots of the MPW and laser voltage in Experiment 2. The MPW value of the uncontrolled sample increased continuously until it exceeded 150 pixels at the end of the fabrication. The MPW value of the conventional PID controlled sample shows a slower growth compared to the uncontrolled sample. Compared to the conventional PID controller, the proposed adaptive control method was able to further stabilize the MPW while keeping it within the range of 115–130 pixels. The proposed adaptive control method was proven effective regardless of the tool path setting.

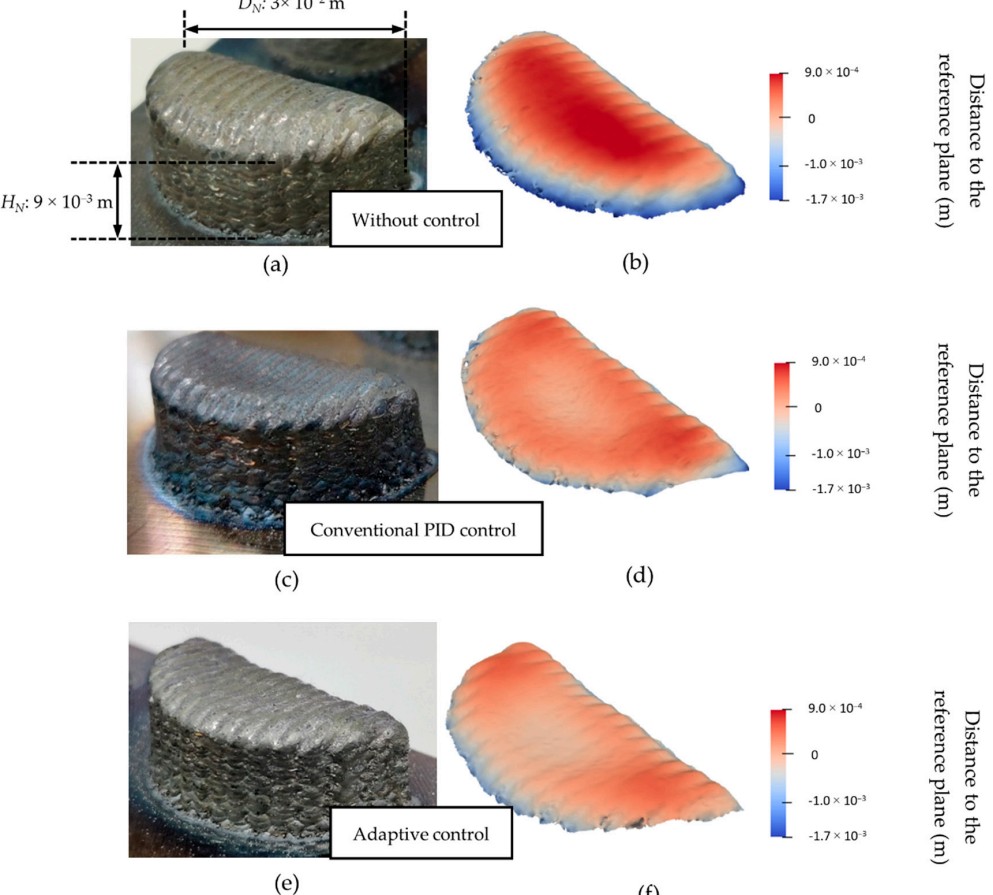

**Figure 8.** Samples of Experiment 2 (solid semicylinder without the profile tool path, 316 L stainless steel). (**a**) The sample fabricated without control; (**b**) The reconstructed surface of the uncontrolled sample; (**c**) The sample fabricated with conventional PID controller; (**d**) The reconstructed surface of the conventional PID controlled sample; (**e**) The sample fabricated with proposed adaptive control method; (**f**) The reconstructed surface of the adaptively controlled sample.

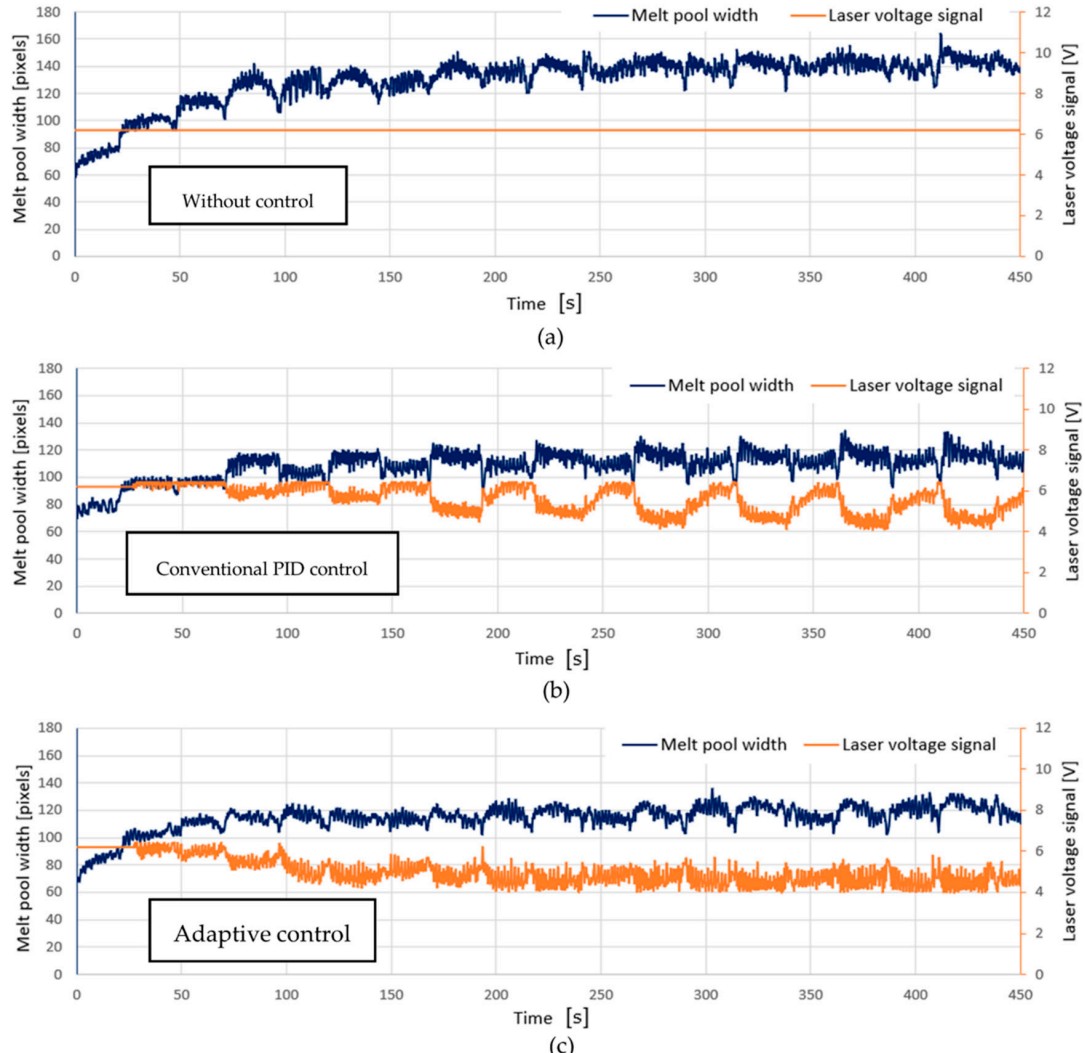

**Figure 9.** Time plots of the MPW and laser voltage signals for Experiment 2 (solid semicylinder without the profile tool path, 316 L stainless steel). (**a**) The data without control; (**b**) The data with conventional PID control; (**c**) The data with the proposed adaptive control.

Experiment 3 was conducted to validate the capability of the proposed adaptive controller in improving the geometric accuracy for any random part, without extra controller tuning or system identification. Compared to the previous two experiments, Experiment 3 involved different material (LPW-35N nickel alloy instead of 316 L stainless steel), geometry (a thin-walled hollow part instead of a solid part), tool path (continuous spiral path instead of zigzag straight path segments), and process parameters (different powder feeding rates and layer thicknesses). The fabrication result of Experiment 3 is shown in Figure 10. Figure 10a shows the uncontrolled sample deposited with the constant laser voltage (6.2 V), which shows a wavy top surface with obvious bulge and dent regions. The highest point and lowest point were measured at 24.25 mm and 21.33 mm, respectively, making a height difference of 2.92 mm. The significant unevenness of the top surface was mainly due to the inconsistent printing velocity along the spiral tool path when the robot carrying the optical head kept accelerating or decelerating in both X and Y directions. The inconsistent velocity resulted in inconsistent laser energy density input to the melt pool. Figure 10b shows the sample fabricated with the proposed adaptive controller enabled. The adaptively controlled sample had a more even surface than the uncontrolled counterparts. The lowest and highest points were 22.90 mm and 24.15 mm, respectively, and the 1.25 mm height difference was less than half that of the uncontrolled sample. As a result of the adaptive closed-loop control, the geometric accuracy of the thin-walled part could be improved. The inconsistent

energy density due to velocity inconsistency was compensated for by the controlled laser voltage. Figure 10c,d shows the time plots of the MPW data and laser voltage signal in Experiment 3. When the MPW rose due to higher energy density (and slower absolute speed), the laser signal was reduced, attempting to lower the MPW, and vice versa. The stabilizing effect of the proposed controller can be observed from the MPW plot in Figure 10d, which has considerably smaller fluctuation than that in Figure 10c. Similar to Experiments 1 and 2, Experiment 3 also demonstrated the capability of the proposed controller in reducing heat accumulation. This capability was more important for thin-walled hollow parts than solid parts since thin-walled parts had smaller cross-sections and hence poorer heat conduction rate. The MPW of the uncontrolled sample grew to nearly 140 pixels in Figure 10c, whereas the MPW of the adaptively controlled sample remained in the narrow range of 95–105 pixels in Figure 10d throughout the process. The reduced heat accumulation due to the relative consistency of the MPW also contributed to the better accuracy of the controlled sample.

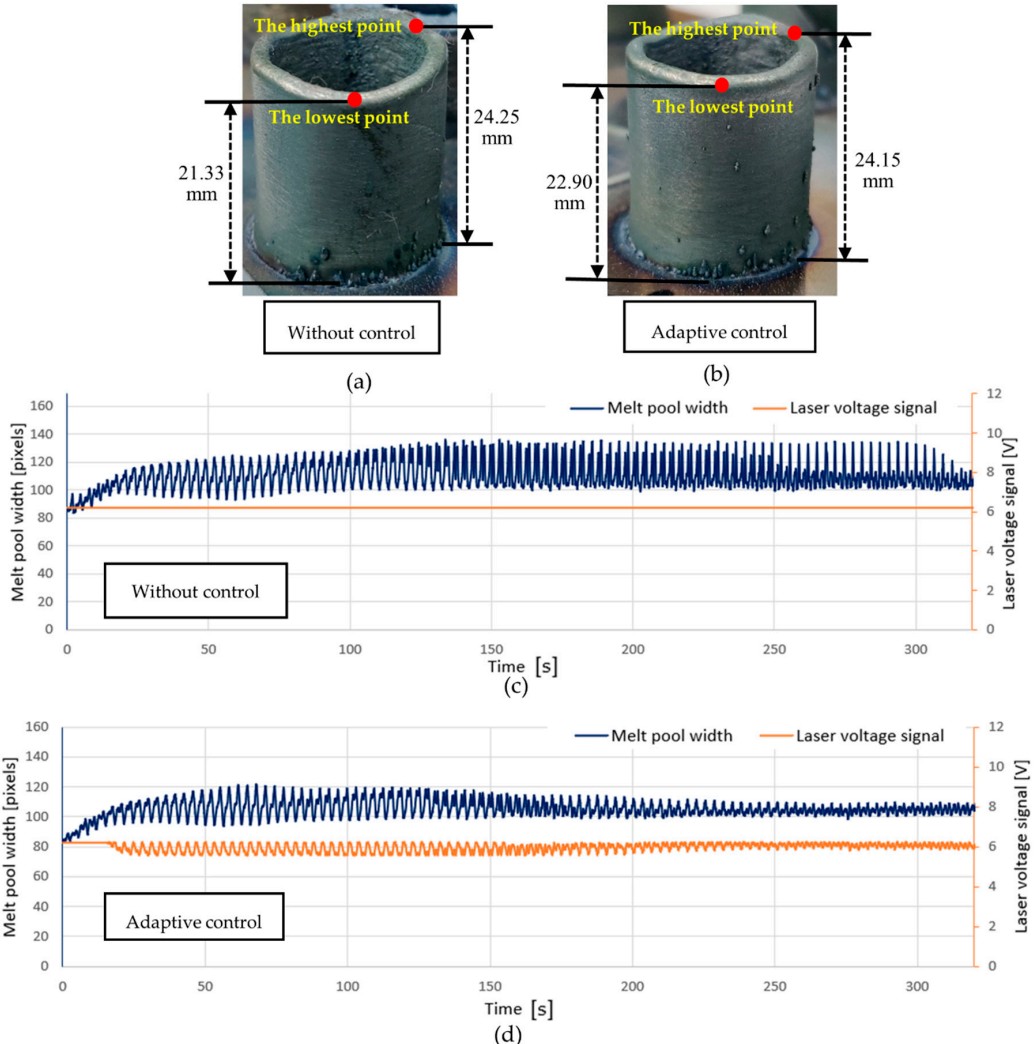

**Figure 10.** Results of Experiment 3 (thin-walled hollow pipes with a spiral tool path, LPW-35N nickel alloy). (**a**) The sample fabricated without control; (**b**) The sample fabricated with adaptive control; (**c**) The MPW and laser voltage plots for the uncontrolled sample; (**d**) The MPW and laser voltage plots for the adaptively controlled sample.

The above experiments demonstrated the effectiveness of the proposed data-driven adaptive control strategy in enhancing the laser-based DED system's performance. In general, the proposed adaptive control method could produce a better result than the uncontrolled and conventional PID controlled processes in terms of MPW stabilization and geometric accuracy improvement. More importantly, the proposed

adaptive control method could eliminate the costly and inefficient controller tuning procedures used in the conventional PID method. A different set of controller parameters needed to be obtained by system identification and trial-and-error experiments in conventional PID whenever the process conditions were changed. In comparison, the proposed data-driven adaptive controller could be used to fabricate parts with any shapes, materials, tool paths, or process parameters. Experiment-based system identification and manual tweaking of the controller was not required even when the DED process conditions were changed for different parts. The controller parameters could be optimized and updated automatically on-the-fly during the DED process without human intervention, and the reduced complexity in controller implementation could pave the way to broader adoption of closed-loop DED systems by industry end-users.

## 5. Conclusions

In this research, a data-driven adaptive control strategy with the automatic parameter tuning capability was proposed for the laser-based DED process. A multitasking controller architecture was developed with the melt pool monitoring unit, autotuning unit, and digital PID unit being executed concurrently. In the autotuning unit, the MPW and laser voltage data were recorded in a temporary buffer periodically before they were used to optimize the controller parameters by the VRFT function. The optimized controller parameters were used to update the digital PID unit automatically. It was demonstrated by experiments that the proposed controller could adapt to different shapes, powder materials, tool paths, and process conditions in DED. Experiments showed improvements in geometric accuracies of DED-fabricated parts as the result of applying the proposed adaptive controller. The improvements were achieved by the melt-pool-stabilizing effect of the controller. The MPW data of controlled samples had less fluctuation and better consistency than those of uncontrolled and conventional PID controlled samples. Another advantage of the proposed controller is that it does not require prior system identification even when the DED process conditions are changed. Controller parameters are updated automatically by the DED process data, and hence experiment-based, layer-dependent, and process-specific control rules are not required. Therefore, the complexity and manpower cost of implementing a closed-loop DED system can be reduced by the proposed method, making it easy for end-users to adopt the controller. The main limitation of this research is that the laser voltage is the only controlled variable in the DED process, while other parameters (e.g., the printing speed and powder feeding rate) are not controlled. In future research, the proposed data-driven adaptive controller can be further developed to take more DED process variables into consideration.

**Author Contributions:** Conceptualization, X.Y. and G.B.; methodology, L.C. and X.Y.; software, L.C. and X.Y.; validation, L.C., X.Y., Y.C. and F.W.; formal analysis, L.C., X.Y., Y.C. and F.W.; writing—L.C., X.Y., S.K.M. and G.B.; writing—review and editing, L.C., X.Y., S.K.M. and G.B.; supervision, S.K.M. and G.B.; project administration, X.Y.; funding acquisition, C.Y. and G.B. All authors have read and agreed to the published version of the manuscript.

**Funding:** This research was funded by A*ccelerate, grant number ACCL/19-GAP077-R20A.

**Acknowledgments:** We acknowledge the support from Nanyang Technological University under the Undergraduate Research Experience on campus (URECA) program.

**Conflicts of Interest:** The authors declare no conflict of interest.

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
