# Peer review of "Data-Driven Adaptive Control for Laser-Based Additive Manufacturing with Automatic Controller Tuning"

_applsci, doi:10.3390/app10227967_

Round 1
Reviewer 1 Report
As the authors have indicated, some researchers have started to focus on the control system for metal AM and the several papers could be found. Compared with these papers, the adaptive control would not be discussed very well by other researchers yet. Therefore, the presented results would gather attention from researchers and engineers who are finding a good control technique for DED or other AMs. If the authors could show remarkable differences from the other studies, I would not hesitate to accept this paper, however, some points should be discussed more. Please check the following points.
General
- The comparison with conventional PIDs
If you want to present the necessity of adaptively controlled PID, the comparison should be performed with the typical PID control, not the constant laser power. In case of your work, the comparison can be conducted easily by keeping the PID gains constant during the DED.
- Evaluation of variations in PID gains
In Fig. 7, you have shown the variations of Kp, Ki, Kd from 150s to 250s. I am sure, however, the drastic variations in Kp, Ki and Kd would be found just after stating the deposition. In other words, you would avoid the discussion of most unstable state in the parameter control, e.g., 0s - 50s. Why don't you discuss the variations in the whole deposition time (0s-450s)?
Small revisions
- In Figs. 1, 5 and 8, the scales of system and deposits are difficult to image. You should add sizes or scale bars for these pictures.
- In Eq.1, kp and ki should be capitalized.
- In Eq.2, the derivative may be expressed with just "s" in the s-domain. You should indicate this is a pseudo differential or the combination of low-pass filter and differential process.
- Eq.15 also should be written in Italics.
- In p.10, l.289, "(M=M+1)" is an expression in programming language. You should say "(M+1)-th cycle" or use any similar expressions.
- In Table 2, the powder supply should be expressed with a unit of "g/min" as many researchers use. You should not express with "rpm."
- According to the results in Fig. 6(b), the laser could not exceed 6.2V under the proposed PID control. Is it the maximum value of laser power (6kV)? If yes, this point also should be explained.
Author Response
Refer to the attached file.

Reviewer 2 Report
- paper that presents a nice topic
- monitoring control is an important subject to control the quality of additive manufacturing components
the authors have set up a methodology and demonstors - concerning the data, rather than giving the speed (rpm) of the powder feeder, it would be more interesting to give the flow (g / min)
- the text on some picture is cut, layout problem or PDF creation?
Author Response
Refer to the attached file.

Round 2
Reviewer 1 Report
I thought that my comment asking to add the experimental results of conventional PID would be too severe to satisfy in short time, but you have done the required experiments sufficiently in high quality and shown an excellent comparison. According to the additional results, the necessity of proposed PID control is more clearly shown than that shown in your original paper. Furthermore, a sufficient explanation is added for the variation in PID gains. I do not hesitate to accept this paper at this time because these two important points were improved enough and the other small points were also sufficiently revised.